# The cation channel TRPA1 tunes mosquito thermotaxis to host temperatures

Román A Corfas[1], Leslie B Vosshall[1,2]*

[1]Laboratory of Neurogenetics and Behavior, The Rockefeller University, New York, United States; [2]Howard Hughes Medical Institute, The Rockefeller University, New York, United States

**Abstract** While most animals thermotax only to regulate their temperature, female mosquitoes are attracted to human body heat during pursuit of a blood meal. Here we elucidate the basic rules of *Aedes aegypti* thermotaxis and test the function of candidate thermoreceptors in this important behavior. We show that host-seeking mosquitoes are maximally attracted to thermal stimuli approximating host body temperatures, seeking relative warmth while avoiding both relative cool and stimuli exceeding host body temperature. We found that the cation channel *TRPA1*, in addition to playing a conserved role in thermoregulation and chemosensation, is required for this specialized host-selective thermotaxis in mosquitoes. During host-seeking, $AaegTRPA1^{-/-}$ mutants failed to avoid stimuli exceeding host temperature, and were unable to discriminate between host-temperature and high-temperature stimuli. *TRPA1*-dependent tuning of thermotaxis is likely critical for mosquitoes host-seeking in a complex thermal environment in which humans are warmer than ambient air, but cooler than surrounding sun-warmed surfaces.

## Introduction

Thermotaxis is a sensory-motor behavior that guides animals toward a preferred temperature. This type of sensory navigation allows animals to avoid environments of noxious cold and heat, with the goal of remaining in physiologically suitable ambient temperatures. For ectotherms, such as most insects, thermotaxis behavior is the primary method of thermoregulation. Terrestrial invertebrates are vulnerable to temperature extremes, facing the risk of desiccation at elevated temperatures, and rapid hypothermia at low temperatures. Therefore, mechanisms to detect environmental temperatures and trigger appropriate approach or avoidance behaviors are extremely important for their survival. For instance, adult *Caenorhabditis elegans* worms migrate preferentially toward a specific thermal environment determined by the temperature of their cultivation (*Hedgecock and Russell, 1975*; *Mori and Ohshima, 1995*). Adult *Drosophila melanogaster* flies prefer a narrow range of air temperatures around 24–25°C (*Sayeed and Benzer, 1996*; *Hamada et al., 2008*) and rapidly avoid air temperatures of ~31°C (*Ni et al., 2013*).

Interestingly, some hematophagous (blood-feeding) arthropods have evolved a specialized mode of thermotaxis to locate endothermic (warm-blooded) hosts. Such thermophilic behavior is seen in kissing bugs [*Triatoma infestans* (*Flores and Lazzari, 1996*) and *Rhodnius prolixus* (*Schmitz et al., 2000*)], the bedbug [*Cimex lectularius* (*Rivnai, 1931*)], the tick [*Ixodes ricinus* (*Lees, 1948*)], and many species of mosquito (*Clements, 1999*) including *Ae. aegypti*, a major tropical disease-vector (*Bhatt et al., 2013*). Female *Ae. aegypti* require a vertebrate blood meal for the production of eggs, and finding a suitable warm-blooded host is therefore an essential component of reproduction. Mosquitoes use a variety of physical and chemical senses to locate hosts in their environment (*Cardé, 2015*). When host-seeking, these animals become strongly attracted to inanimate warm objects, eagerly probing at them as if they were hosts (*Howlett, 1910*).

*For correspondence: Leslie.Vosshall@rockefeller.edu

Competing interests: The authors declare that no competing interests exist.

**eLife digest** Temperature can vary considerably in an environment. Living organisms have evolved sensory systems to detect and avoid excessive heat or cold: a behavior that is termed 'thermotaxis'. In rare cases, animals use this ability to locate food sources in their environment. One example of such an adaptation is the female mosquito of the species *Aedes aegypti*. When a mosquito needs blood to produce her eggs, she becomes attracted to the body heat of warm-blooded hosts. But the range of temperatures that these mosquitoes prefer and the genes required for this behavior had not been been defined.

Now, Corfas and Vosshall have found that female *Aedes aegypti* are highly sensitive to differences in temperature, and are capable of heat-seeking in a range of environmental temperatures. Furthermore, by seeking out things that are warmer than their surroundings, while avoiding those that are cooler or much hotter than their host's body temperatures, these mosquitoes tune their thermotaxis toward targets that resemble a human to feed upon.

Corfas and Vosshall also discovered that a protein called TRPA1 is required for this tuning of *Aedes aegypti*'s heat-seeking behavior. This protein is known to allow insects to detect chemical signals and regulate their own temperature, but it was not previously known that this protein was involved in mosquito thermotaxis. Mutant mosquitoes without the gene for TRPA1 failed to avoid high temperatures, which meant that they could no longer tell the difference between an overly hot target and a warm one that resembled their hosts.

Following on from this work, the next challenge will be to characterize all the genes, sensory organs, and neural circuits that drive mosquito heat-seeking behavior. These findings may in the future inform the design of the next generation of repellents and traps for the control of mosquito-borne diseases, such as dengue and yellow fever.

In nature, mosquitoes thermotax in a complex thermal landscape in which ambient air temperature, host body temperature, and surrounding surface temperatures can vary widely. For mosquitoes such as *Ae. aegypti*, host-seeking behavior can be activated by an increase in ambient carbon dioxide ($CO_2$) (**Majeed et al., 2014**). This activation elicits flight activity (**Eiras and Jepson, 1991**; **McMeniman et al., 2014**) and results in an array of behaviors including attraction to visual stimuli (**van Breugel et al., 2015**) and host olfactory cues (**Dekker et al., 2005**; **McMeniman et al., 2014**), and landing on warm objects (**Burgess, 1959**; **Eiras and Jepson, 1994**; **Kröber et al., 2010**; **Maekawa et al., 2011**; **McMeniman et al., 2014**; **van Breugel et al., 2015**). *Ae. aegypti* flying in a wind tunnel can detect a warmed stimulus from a distance, eliciting attraction and thermotaxis (**van Breugel et al., 2015**).

What are the mechanisms by which animals detect thermal stimuli, and how might these be adapted for the specialized needs of heat-seeking female mosquitoes? Thermotaxis is typically initiated by thermosensitive neurons that sample environmental temperature to inform navigational decision-making. Such neurons must be equipped with molecular thermosensors capable of detecting and transducing thermal stimuli. Diverse molecular thermoreceptors have been identified in the animal kingdom, many of which are members of the transient receptor potential (TRP) superfamily of ion channels (**Barbagallo and Garrity, 2015**; **Palkar et al., 2015**). Different thermosensitive TRPs show distinct tuning spanning the thermal spectrum from noxious cold to noxious heat. Among these is TRPA1, which is a heat sensor in multiple insects, including the vinegar fly *D. melanogaster* and the malaria mosquito *Anopheles gambiae* (**Hamada et al., 2008**; **Wang et al., 2009**). Neurons in thermosensitive sensilla (**Gingl et al., 2005**) of *An. gambiae* female antennae express *TRPA1* (**Wang et al., 2009**). In *D. melanogaster*, *TRPA1* is expressed in internal thermosensors located in the brain, and *DmelTRPA1[-/-]* mutants fail to avoid high air temperature in a thermal gradient (**Hamada et al., 2008**). Interestingly, some snakes and vampire bats express thermosensitive TRP channels in organs used to sense infrared radiation from warm-blooded prey (**Gracheva et al., 2010**; **2011**). This raises the possibility that *AaegTRPA1* may be used by mosquitoes to find hosts. Recently, a structurally distinct insect thermosensor, *Gr28b*, was identified in *D. melanogaster* (**Ni et al., 2013**). *Gr28b*, a gustatory receptor paralog, is expressed in heat-sensitive neurons of *D. melanogaster* aristae and is an important component of thermotaxis during rapid avoidance of heat

(*Ni et al., 2013*). It is also highly conserved among *Drosophila* species (*McBride et al., 2007*), and has a clear ortholog in *Ae. aegypti, AaegGr19* (*Ni et al., 2013*). A functional role for these thermo-sensors has never been investigated in the mosquito.

Here, we use high-resolution quantitative assays to examine the behavioral strategies underlying mosquito heat-seeking behavior. Our results show that by seeking relative warmth and avoiding both relative cool and high temperatures, female mosquitoes selectively localized to thermal stimuli that approximate warm-blooded hosts. Using genome editing, we generated mutations in the candidate thermoreceptors, *AaegTRPA1* and *AaegGr19*. We found that *TRPA1* is required for tuning mosquito thermotaxis during host-seeking. *AaegTRPA1$^{-/-}$* mutants lacked normal avoidance of thermal stimuli exceeding host body temperatures, resulting in a loss of preference for biologically relevant thermal stimuli that resemble hosts. This work is important because it identifies a key mechanism by which mosquitoes tune their thermosensory systems toward human body temperatures.

## Results

We previously described an assay to model heat-seeking behavior in the laboratory by monitoring mosquitoes landing on a warmed Peltier element in the context of a cage supplemented with $CO_2$ (*Figure 1A,B*) (*McMeniman et al., 2014*). This assay has the advantages that it is simple in design, produces robust behaviors, and enables the collection of data from large numbers of animals in a short experimental timeframe. Using this system, we can examine mosquito responses to diverse thermal stimuli and measure thermotaxis in different ambient temperature environments. We first needed to determine whether heat-seeking behavior habituates over multiple thermal stimulations. In our heat-seeking assay, *Ae. aegypti* mosquitoes reliably responded to 12 serial presentations of a 3-minute long 40°C stimulus over the course of more than 2.5 hr, with no evidence of habituation (*Figure 1—figure supplement 1*).

*Ae. aegypti* can feed on a variety of hosts (*Clements, 1999*; *Tandon and Ray, 2000*) with core body temperatures ranging from ~37°C (humans) to ~40–43°C (chickens) (*Richards, 1971*) (*Figure 1C*). It is unknown whether there are minimal or maximal temperature thresholds constraining mosquito heat-seeking, and whether responses to thermal stimuli depend on the background ambient temperature.

To investigate these questions, we measured attraction to thermal stimuli produced by heating the Peltier to temperatures ranging from ambient (set to 26°C in these experiments) to 60°C (*Figure 1D, E*). We found that mosquitoes were highly sensitive to thermal contrast and were attracted to stimuli 2.5°C above ambient (*Figure 1D–F*). Furthermore, mosquito occupancy on the Peltier increased monotonically with stimulus temperatures up to 40°C. However, for higher temperature stimuli, we observed a dramatic reduction in Peltier occupancy. A 50°C stimulus resulted in approximately half as many animals on the Peltier compared to a 40°C stimulus (*Figure 1D–F*). Stimuli of 55°C or greater resulted in occupancy rates indistinguishable from an ambient thermal stimulus (26°C) (*Figure 1D–F*). Spatial analysis of mosquito occupancy on or near the Peltier revealed that while mosquitoes were still attracted to high-temperature stimuli, they populated the area peripheral to the Peltier, and strongly avoided the Peltier itself for stimuli $\geq$ 55°C (*Figure 1F*).

Female mosquitoes searching for a warm-blooded host may be responding to the absolute temperature of a stimulus or may instead be evaluating relative warmth, defined as the differential between a stimulus and background ambient temperature. To investigate the thermotaxis strategies constituting mosquito heat-seeking behavior, we conducted experiments at three ambient temperatures: 21, 26, and 31°C (*Figure 2A–F*). We found that Peltier occupancy for stimuli 21–40°C depended on the differential between the Peltier and ambient temperature (*Figures 2B,C*), rather than the absolute temperature of the Peltier (*Figure 2A*). For example, at all ambient temperatures tested, a stimulus 5°C above ambient was sufficient to elicit significant heat-seeking, and elicited approximately half as much Peltier occupancy as a stimulus 10°C above ambient. On the other hand, heat-seeking to targets 50–55°C was inhibited at all ambient temperatures tested (*Figures 2D,F*), despite the fact that the temperature differential varied widely in these situations (*Figure 2E*).

These results show that *Ae. aegypti* thermotaxis is driven by seeking relative warmth, but restricted by an absolute upper threshold of ~50–55°C. Because female mosquitoes are attracted to relative warmth, we hypothesized that they may also avoid relative cool. This complementary

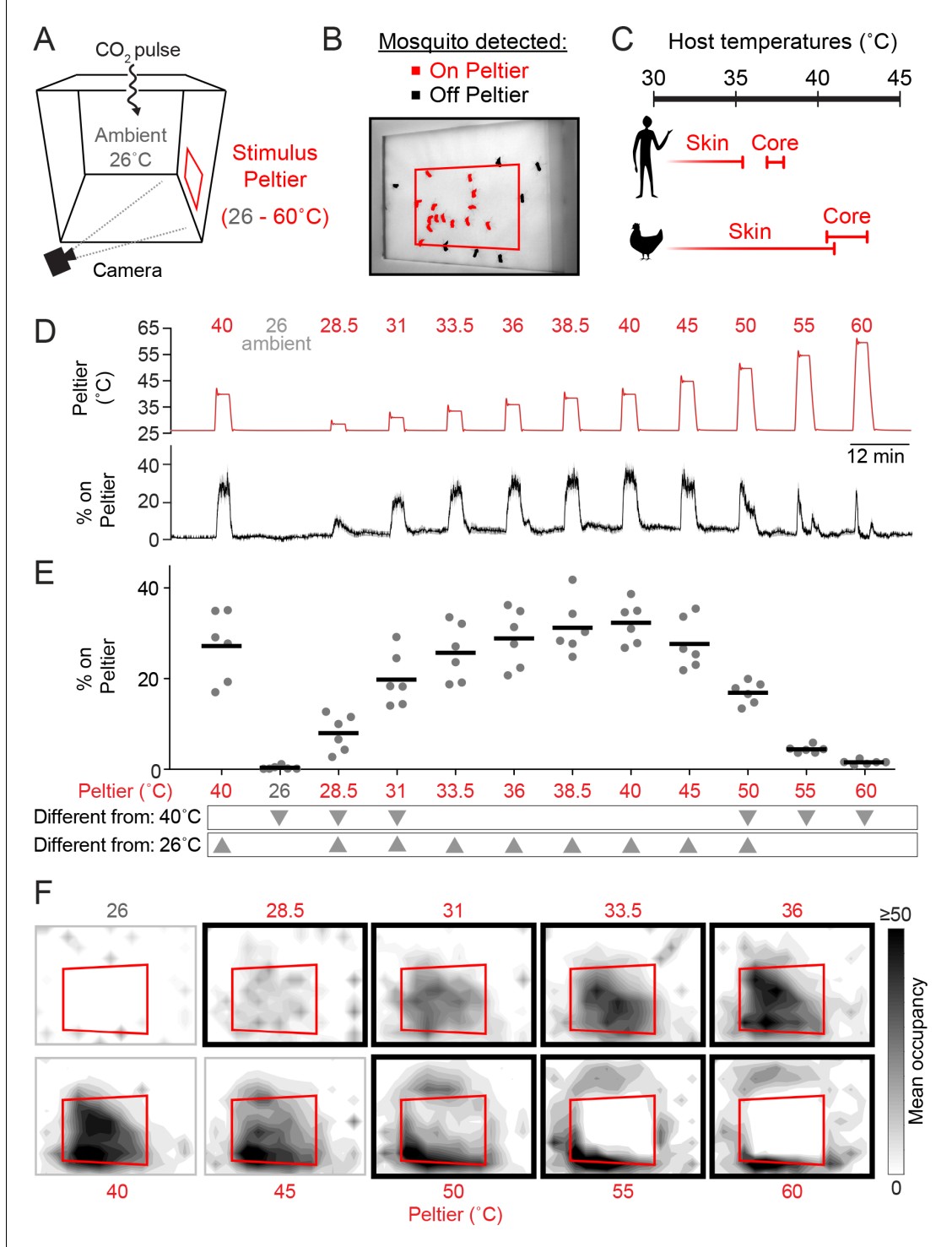

**Figure 1.** Mosquitoes thermotax to stimuli approximating host body temperature. (**A**) Schematic of heat-seeking assay enclosure (30 × 30 × 30 cm). (**B**) Representative experimental image showing mosquitoes detected on and near the Peltier (red square). (**C**) Typical skin and core temperatures of *Ae. aegypti* hosts, humans and chickens (*Richards, 1971*; *Yao, et al., 2008*. (**D–F**) Heat-seeking behavior measured for a range of stimuli from 26 to 60°C (n = 6 trials per condition). Peltier temperature measured by thermocouple (**D**, top trace, mean in red, s.e.m. in gray) and percent of mosquitoes on Peltier (**D**, bottom trace, mean in black, s.e.m. in gray). We note that variance in both traces is low, making s.e.m. traces difficult to see. (**E**) Percent mosquitoes on Peltier during seconds 90–180 of each stimulus period in (**D**). Each replicate is indicated by a dot, and mean by a line. Arrowheads indicate significant differences (p < 0.05) from the second presentation of the 40°C stimulus or from 26°C (repeated measures one-way ANOVA with Bonferroni correction). (**F**) Heat maps showing mean mosquito occupancy on the Peltier (red square) and surrounding area, during seconds 90–180 of each

*Figure 1 continued on next page*

*Figure 1 continued*

stimulus period in (D). Bold borders indicate stimuli with responses significantly different from 26°C stimulus (top row) or 40°C stimulus (bottom row) in (E) (p < 0.05; repeated-measures ANOVA with Bonferroni correction).

The following figure supplement is available for figure 1:

**Figure supplement 1.** Mosquitoes consistently thermotax to repeated 40°C stimuli.

behavior would serve to improve host-seeking thermotaxis. We examined mosquito responses to cooling by analyzing the rate at which animals left the Peltier when it cooled at the conclusion of a stimulus period. We found that mosquitoes left the Peltier at similar rates regardless of the absolute temperature of the stimulus (*Figure 2G,H*, *Figure 2—figure supplement 1*; based on analysis of data in *Figure 1D,E*), demonstrating that mosquitoes avoid relative cool during heat-seeking.

Our characterization of *Ae. aegypti* heat-seeking revealed multiple behavioral components contributing to selective thermotaxis during host-seeking: 1) the seeking of relative warmth; 2) the avoidance of relative cool; and 3) the avoidance of thermal stimuli exceeding host temperature. Each of these sensory-motor functions may rely on the same molecular thermosensors, or may instead use distinct thermosensors. We considered the possibility that thermoreceptors ordinarily dedicated to the behavioral thermoregulation typical of most ectotherms such as *D. melanogaster*, may have evolved a function in host-seeking by mosquitoes and other hematophagous arthropods. Using this reasoning, we generated *AaegTRPA1*$^{-/-}$ mutants using zinc-finger nuclease-mediated genome editing (*Figure 3A*, *Figure 3—figure supplement 1A*).

In addition to its function as a thermoreceptor, TRPA1 is a highly conserved chemosensor of electrophile irritants such as N-methylmaleimide (*Macpherson et al., 2007*; *Kang et al., 2010*). Using a modified capillary feeding (CAFE) assay (*Ja et al., 2007*) (*Figure 3B*), we found that wild-type *Ae. aegypti* mosquitoes strongly avoided consumption of N-methylmaleimide (*Figure 3C*), as well as the bitter compound denatonium benzoate (*Figure 3D*). *AaegTRPA1*$^{-/-}$ mutants rejected denatonium benzoate (*Figure 3D*) but did not avoid consumption of N-methylmaleimide (*Figure 3C*). We interpret this result as a loss of N-methylmaleimide detection in *AaegTRPA1*$^{-/-}$ mutants, leading to no preference between sucrose and sucrose containing N-methylmaleimide. We note that this simple CAFE assay could be used to discover additional mosquito anti-feedants to repel *Ae. aegypti*, beyond the two chemicals identified here.

Because *TRPA1* is important in insect thermoregulation (*Hamada et al., 2008*), we used a modified thermal gradient assay (*Sayeed and Benzer, 1996*; *Hamada et al., 2008*) to assess thermal preference in wild-type and *AaegTRPA1*$^{-/-}$ mutant mosquitoes (*Figure 3—figure supplement 1B*). *AaegTRPA1*$^{-/-}$ mutants were impaired in avoidance of high air temperature, leading to significant mortality (*Figure 3—figure supplement 1C–G*). Together, these data indicate that TRPA1 has a conserved chemosensory and thermosensory function in *Ae. aegypti*.

We next asked if *AaegTRPA1* is required for mosquito heat-seeking behavior. *AaegTRPA1*$^{-/-}$ mutants showed normal attraction to stimuli at or below 45°C, but strikingly lacked normal avoidance of higher temperature stimuli (50°C and 55°C) (*Figures 3E,F*). A detailed analysis of Peltier occupancy over time revealed that *AaegTRPA1*$^{-/-}$ mutants persisted on the Peltier during 50, 55, and 60°C stimulus presentations, whereas control animals rapidly left these high-temperature stimuli (*Figure 3G*). Therefore, *AaegTRPA1* is not required for initial attraction to warmth but is required for normal avoidance of high-temperature stimuli that exceed host body temperature.

Because *AaegTRPA1*$^{-/-}$ mutants retained attraction to warm stimuli, we used targeted mutagenesis to test a requirement for *AaegGr19*, the *Ae. aegypti* ortholog of *Gr28b*, in heat-seeking (*Figure 3—figure supplement 2A,B*). Although this thermosensor is required for mediating rapid avoidance of warmth in *D. melanogaster* (*Ni et al., 2013*), *AaegGr19*$^{-/-}$ mutant mosquitoes showed no thermotaxis defects (*Figure 3—figure supplement 2C*). The different phenotypes of these mutations in *Ae. aegypti* and *D. melanogaster* may reflect differences in expression patterns of these genes. *DmelTRPA1* is expressed in internal thermosensors of the brain (*Hamada et al., 2008*), while thermosensitive isoforms of *DmelGr28b* are expressed in peripheral heat-sensors (*Ni et al., 2013*). In *Ae. aegypti*, *TRPA1* RNA is expressed in numerous tissues including the antennae (*Matthews et al., 2015*), where it may be associated with peripheral thermosensitive sensilla responding to rapid

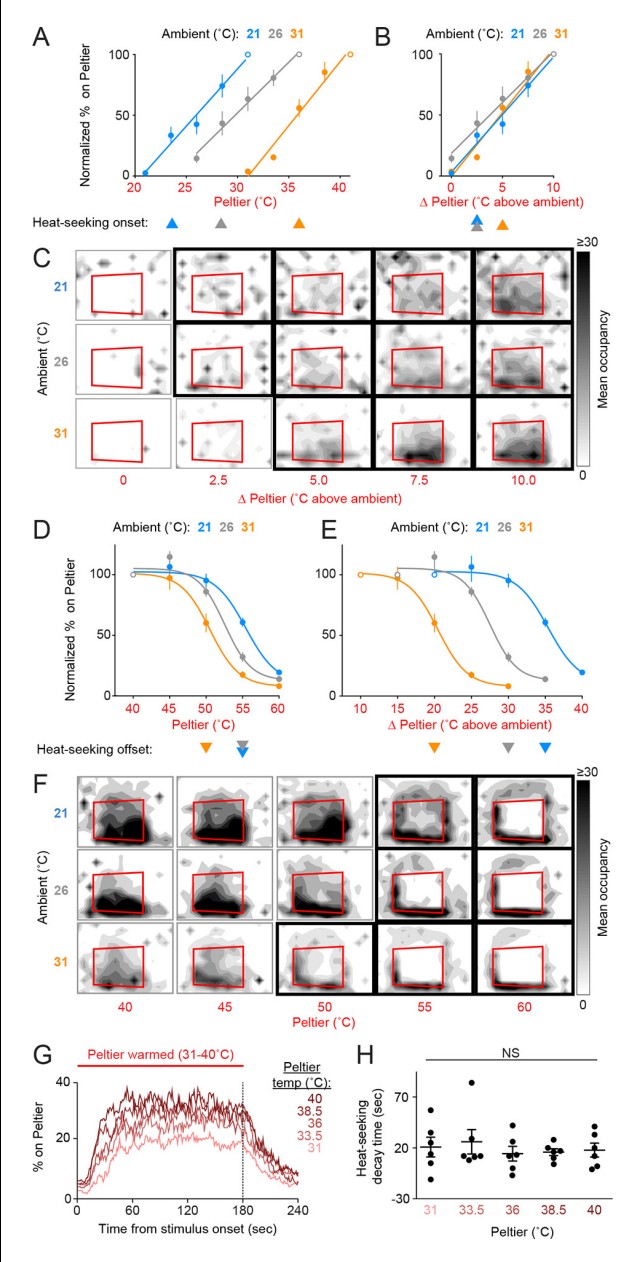

**Figure 2.** Mosquitoes thermotax to relative warmth and avoid both relative cooling and stimuli exceeding host body temperature. (A–F) Heat-seeking at different ambient temperatures (n = 5–6 trials per condition): 21°C (blue), 26°C (gray), 31°C (orange). Data in **A, B, D,** and **E** are plotted as mean ± s.e.m. (**A,D**) Percent of mosquitoes on Peltier during seconds 90–180 of stimuli of indicated temperature, normalized to stimulus 10°C above ambient (**A,** open circle) or 40°C stimulus (**D,** open circle). (**B,E**) Same data as in (**A**) and (**D**), respectively, plotted using differential between ambient and Peltier temperature. For each ambient temperature, arrowheads indicate the lowest temperature stimulus found to elicit a significant increase in heat-seeking compared to an ambient temperature stimulus (**A,B**) or a reduction in heat-seeking compared to a 40°C stimulus (**D,E**) (p < 0.05; repeated-measures ANOVA with Bonferroni correction). For each ambient temperature, linear regressions (**A, B,** 21°C: 10.6/° C, $R^2$ = 0.98, 26°C: 12/°C, $R^2$ = 0.99, 31°C: 9.5/°C, $R^2$ = 0.97) or variable slope sigmoidal dose–response curves (**D, E,** 21°C: $IC_{50}$ = 55.4°C, $R^2$ = 0.87, 26°C: $IC_{50}$ = 52.5°C, $R^2$ = 0.92, 31°C: $IC_{50}$ = 50.5°C, $R^2$ = 0.91) are plotted. (**C,F**) Heat maps showing mean mosquito occupancy on the Peltier (red square) and surrounding area, during seconds 90–180 of each stimulus period. Bold borders indicate stimuli with responses significantly different from an ambient-temperature stimulus (**C**) in (**A,B**), or significantly different from a 40°C stimulus (**F**) in (**D,E**) (p < 0.05; repeated-measures ANOVA with Bonferroni correction). (**G,H**) Analysis of mosquito responses to cooling from

*Figure 2 continued on next page*

*Figure 2 continued*
data in (*Figure 1D*). (G) Mean percent of mosquitoes on Peltier during thermal stimuli 31–40℃. Dashed line indicates the end of the stimulus period. (H) Post-stimulus time at which the percent of mosquitoes on Peltier has decayed to one half of the mean during seconds 90–180 of the stimulus period from (*Figure 1E*). Each replicate is indicated by a dot, mean ± s.e.m. by lines (NS, not significant; one-way ANOVA with Bonferroni correction).
The following figure supplement is available for figure 2:

**Figure supplement 1.** Dynamics of Peltier temperature during stimulus periods.

thermal fluctuations (*Gingl et al., 2005*), as it is in *An. gambiae* (*Wang et al., 2009*). The cellular expression pattern of *Gr19* in *Ae. aegypti* is not known, but its transcript is broadly expressed (*Matthews et al., 2015*).

Although *AaegTRPA1^-/-^* mutants did not show normal avoidance of high-temperature stimuli, they may still prefer host-temperature stimuli if presented with a choice. In a heat-seeking choice assay with two independently controlled Peltiers, we examined the importance of *AaegTRPA1* in guiding mosquito thermotaxis in a more complex thermal landscape (*Figure 4A*). In this assay, mosquitoes were simultaneously presented with two thermal stimuli. When presented with two 40℃ stimuli, both wild-type and mutant mosquitoes distributed equally between the Peltiers (*Figure 4B*), but in a choice between a 40℃ and 50℃ stimulus, wild-type mosquitoes strongly preferred the 40℃ Peltier (*Figure 4C*) and avoided the 50℃ Peltier (*Figure 4D*). Remarkably, in this choice scenario, *AaegTRPA1^-/-^* mutants failed to avoid the 50℃ Peltier, resulting in no preference for the 40℃ stimulus (*Figure 4 C, D*, *Video 1*).

## Discussion

We have elucidated the basic thermotaxis strategies used by mosquitoes, and revealed an important role for *TRPA1* in regulating this behavior (*Figure 4E*). Using a quantitative thermotaxis assay, we modelled *Ae. aegypti* heat-seeking behavior in the laboratory. We found that mosquitoes can search for hosts in a wide range of ambient temperatures by seeking relative warmth and avoiding relative cool. Remarkably, these animals can detect a stimulus with thermal contrast as small as 2.5℃. In an outdoor environment, however, hosts are often warmer than the surrounding air but cooler than sun-warmed soil, rocks, trees, and human-made objects (*Figure 4F*). For this reason, diurnal mosquitoes such as *Ae. aegypti* are poorly served by merely thermotaxing to the hottest object available. A more optimal strategy is to search specifically for biologically relevant stimuli, and to avoid thermal stimuli exceeding host temperature, as we have observed in our laboratory models of heat-seeking. Acquiring a blood meal is an essential component of reproduction for a female *Ae. aegypti* mosquito. To maximize her chances of success, a female mosquito should reject 'distracting' stimuli that exceed host temperatures. Our results demonstrate that *AaegTRPA1* is critical for this selective thermotaxis.

Mosquito heat-seeking behavior represents an excellent model system for further study of the genetics (*Kang et al., 2012*; *Zhong et al., 2012*), neuroscience (*Frank et al., 2015*; *Liu et al., 2015*), and decision-making (*Luo et al., 2010*) underlying thermosensation and thermotaxis. Until now, mechanistic studies of thermosensation have been largely restricted to traditional laboratory model organisms, such as domestic mice and *Drosophila melanogaster* flies, whose thermotaxis consists mainly of moving away from suboptimal thermal environments. Mosquitoes too, must undergo such behavioral thermoregulation, as we have found in our thermal gradient assay. However, their repertoire of thermotactic behaviors is expanded by the evolution of a specialized and highly tuned mode of thermotaxis to locate warm-blooded hosts. It may also be that these two thermotactic drives—host-seeking and thermoregulation—interact in mosquitoes. For instance, the avoidance of high-temperature stimuli during heat-seeking may be influenced by a nociceptive or thermoregulatory response independent of host-seeking behavior. It will be interesting to investigate the neural mechanisms that regulate divergent behavioral choices of thermoregulation and heat-seeking, and whether these systems are in behavioral conflict during mosquito host-seeking.

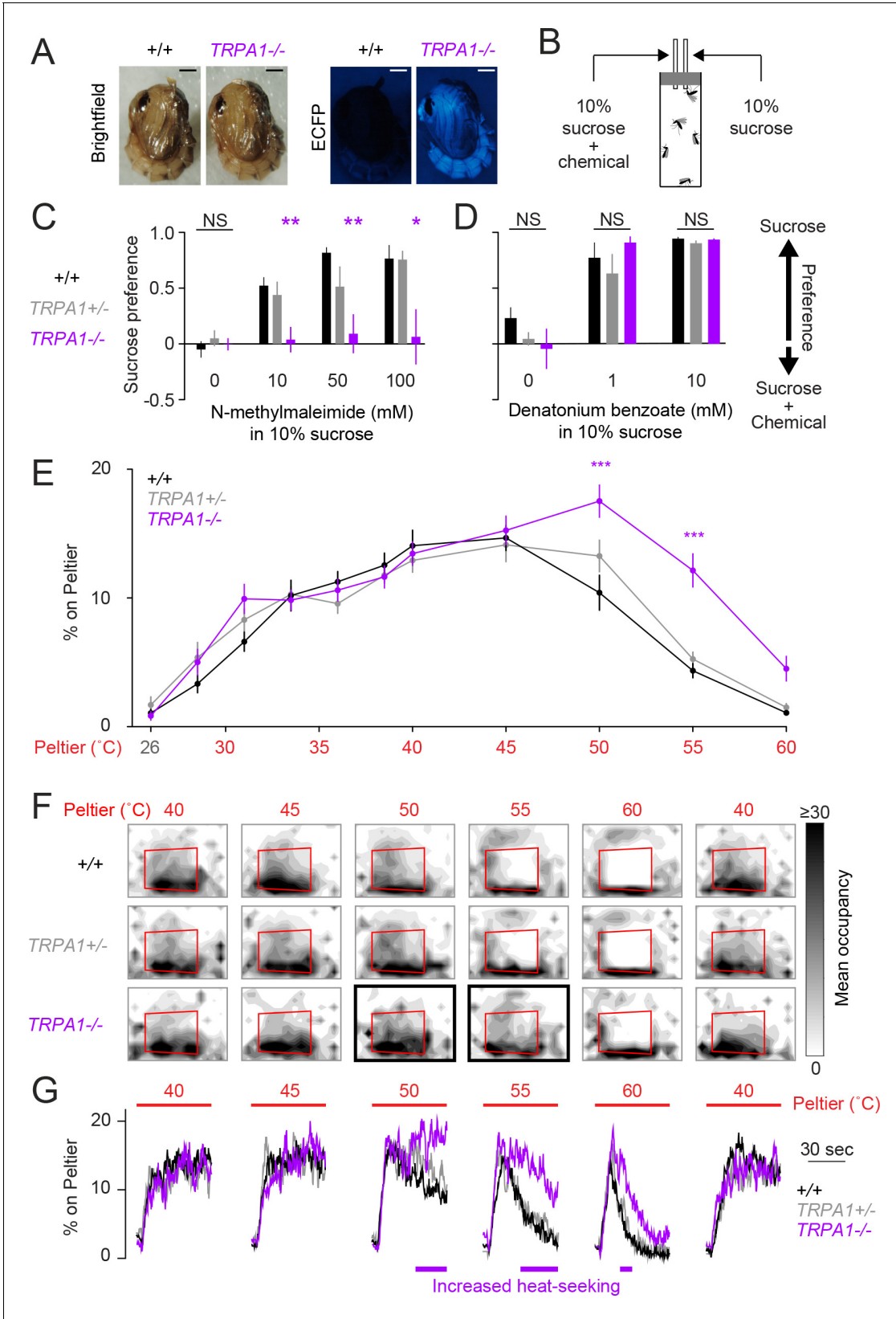

**Figure 3.** *AaegTRPA1⁻/⁻* mutants fail to avoid a chemical irritant and high-temperature stimuli. (**A**) Representative bright field (left) and fluorescence (right) images of wild-type and *AaegTRPA1⁻/⁻* female pupae marked by ubiquitous expression of enhanced cyan fluorescent protein (ECFP). Scale bars: 0.5 mm. (**B**) Schematic of capillary feeding (CAFE) assay. (**C,D**) Sucrose preference over sucrose containing the indicated concentration of N-

*Figure 3 continued on next page*

*Figure 3 continued*

methylmaleimide (**C**, n = 10–12 trials per condition) or denatonium benzoate (**D**, n = 7 trials per condition) for mosquitoes of the indicated genotypes (NS, not significant; *p < 0.05, **p < 0.01; one-way ANOVA with Bonferroni correction compared to wild-type). (**E**) Percent of mosquitoes of indicated genotypes on Peltier during seconds 90–180 of stimuli of indicated temperature (mean ± s.e.m., n = 6–9 trials per genotype; ***p < 0.001; repeated measures one-way ANOVA with Bonferroni correction). (**F**) Heat maps showing mean mosquito occupancy for the indicated genotypes on the Peltier (red square) and surrounding area, during seconds 90–180 of each stimulus period. Bold borders indicate stimuli with responses significantly different from wild-type in (**E**) (p < 0.05; repeated-measures ANOVA with Bonferroni correction). (**G**) Mean percent of mosquitoes of indicated genotypes on Peltier during seconds 0–180 of thermal stimuli 40–60°C and during subsequent re-presentation of 40°C. Timespans with statistically significant increases in *AaegTRPA1*<sup>-/-</sup> mutant Peltier occupancy compared to wild-type are indicated by purple lines (calculated from 15 second bins; p < 0.05; one-way ANOVA with Bonferroni correction).

The following figure supplements are available for figure 3:

**Figure supplement 1.** *AaegTRPA1*<sup>-/-</sup> mutants fail to avoid noxious heat in a thermal gradient.

**Figure supplement 2.** *AaegGr19*<sup>-/-</sup> mutants show normal thermotaxis.

Our work identifies *TRPA1* as a gene regulating mosquito avoidance of high-temperature stimuli, which we have shown to be a major behavioral component of heat-seeking. However, because both *AaegTRPA1*<sup>-/-</sup> and *AaegGr19*<sup>-/-</sup> mutants retain normal attraction to warmth, this aspect of heat-seeking must rely on other thermoreceptors, still to be identified. Our study shows that this attraction must be mediated either by a single thermosensor that adapts to background temperature, or multiple thermosensors each tuned to a distinct absolute threshold. During interactions with a warm-blooded host or a 50°C Peltier, mosquitoes are likely experiencing a wide range of thermal fluctuations. For instance, there can be a 10°C thermal gradient in the air within 5 mm of a 37°C stimulus or human arm (*van Breugel et al., 2015*). This suggests that for a mosquito standing on a warmed Peltier, the antennae, brain, thorax, forelegs, and proboscis may all be experiencing different temperatures. The temperature of mosquito tissues will also be greatly influenced by their material and geometric properties, as well as thermal conduction due to contact with the Peltier. Furthermore, convective plumes forming in the air near a vertical heated plate can be highly dynamic and turbulent in their structure, with thermal air gradients differing considerably at the bottom and top of the plate (*Bejan, 2013*). These features of thermal stimuli may explain why mosquitoes in our assay often appear to be differentially attracted to the bottom and top of the Peltier. Future studies characterizing this complex thermal microenvironment, and identifying the relevant thermosensory neurons and receptors will be required to define the thermal fluctuations experienced by mosquitoes during heat-seeking. Understanding the behavioral and molecular basis of thermotaxis in mosquitoes and other disease vectors (*Flores and Lazzari, 1996*; *Schmitz et al., 2000*) is of great biomedical importance. *Ae. aegypti* mosquitoes are potent vectors of yellow fever, chikungunya, and dengue arboviruses, resulting annually in hundreds of millions of infections (*Bhatt et al., 2013*). Further study of mosquito heat-seeking behavior may aid in the design of next-generation traps, repellents, and control strategies.

## Materials and methods

### Mosquito rearing and maintenance

*Ae. aegypti* wild-type (*Orlando*), *AaegGr19*, and *AaegTRPA1* mutant strains were maintained and reared at 25–28°C, 70–80% relative humidity with a photoperiod of 14 hr light:10 hr dark (lights on at 8 a.m.) as previously described (*DeGennaro et al., 2013*). Adult mosquitoes were provided constant access to 10% sucrose solution for feeding, and females were provided with a blood source for egg production, either live mice or human volunteers. Blood-feeding procedures were approved and monitored by The Rockefeller University Institutional Animal Care and Use Committee and Institutional Review Board, protocols 14756 and LV-0652 respectively. Human volunteers gave their informed written consent to participate in mosquito blood-feeding procedures. Before behavioral assays, mosquitoes were sexed and sorted under cold anesthesia (4°C) and fasted for 15–24 hr in the presence of a water source.

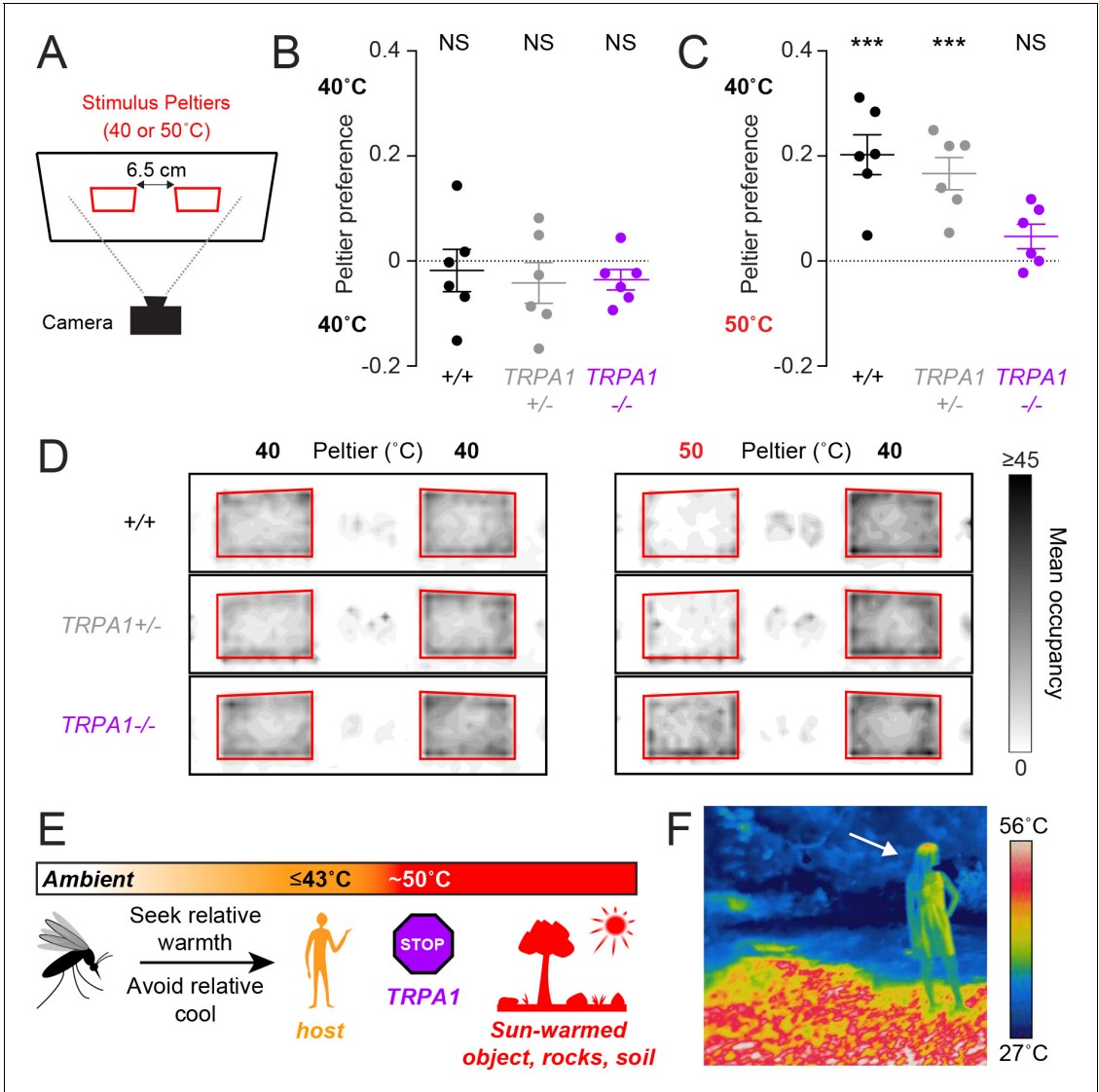

**Figure 4.** *AaegTRPA1−/−* mutants fail to discriminate between host-temperature and higher-temperature targets. (**A**) Schematic of heat-seeking choice assay. (**B,C**) Preference for 40°C versus 40°C (**B**) or 50°C versus 40°C (**C**) Peltiers for indicated genotypes (n = 6 trials per genotype; mean ± s.e.m., with each replicate indicated by a dot; NS, not significant; ***p < 0.001; one sample t-test versus zero preference). In (**C**) *AaegTRPA1−/−* mutants are significantly different from wild-type and heterozygous mutants (p < 0.05, one-way ANOVA with Bonferroni correction). (**D**) Heat maps showing mean mosquito occupancy for the indicated genotypes on Peltiers (red squares) of the indicated temperatures and surrounding area, during seconds 60–240 of each stimulus period. (**E**) Model of mosquito thermotaxis. (**F**) Thermal image of a person (arrow) standing on a sunlit patch of grass in Central Park in New York City.

## ZFN-mediated targeted mutagenesis

### Molecular biology

PCR was carried out using Novagen KOD polymerase (EMD Millipore, Billerica, MA), products were cloned using pCR4-TOPO (Invitrogen), and Sanger sequenced by Genewiz (South Plainfield, NJ).

### ZFN design

ZFNs targeting *AaegTRPA1* or *AaegGr19* (VectorBase Accession numbers AAEL009419 and AAEL011073, respectively) were designed and produced by the CompoZr Custom ZFN Service (Sigma-Aldrich, St. Louis, MO). The nucleotide sequences of the ZFN-binding sites (upper case) and nonspecific cut site for wild-type heterodimeric Fok1 endonuclease (lower case) are:

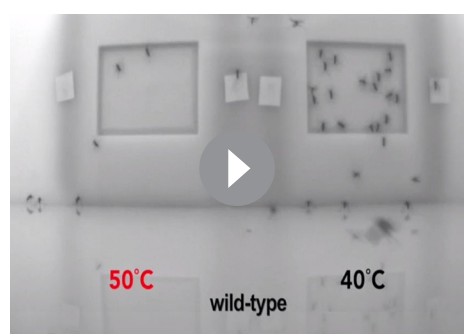

**Video 1.** *AaegTRPA1* is required for tuning avoidance of high-temperature stimuli during heat-seeking. *AaegTRPA1*$^{-/-}$ mutants presented with a choice between two Peltiers, one at 40°C and one at 50°C. Video is sped up 10-fold (images acquired at 1 Hz and reproduced at 10 frames/s), and shows seconds 60–240 of the stimulus period.

*AaegTRPA1*: 5'-GTCGTTTTCGTCCATAC-CgatgtcGTTGCTTAGGACGTT-3'.
*AaegGr19*: 5'-ACCAACCT-TTCACTGCaaatgacCCACCGGAAAGTGGCA-3'.

## Homologous recombination design

Donor plasmids were generated as previously described (*McMeniman et al., 2014*). All donor plasmids included homologous arms cloned from wild-type genomic DNA, and consisting of sequence flanking or partially overlapping the ZFN recognition sites.

*AaegTRPA1* was targeted using *pSL1180-HR-PUbECFP-TRPA1*. The left homologous arm (1608 bp, primers: forward, 5'-GCATGCATGGG-TAACAAGAAGGGTTGT-3' and reverse, 5'-CGA-CAAGTGGTTTACTTGTGTGTCAATC-3') and right homologous arm (3037 bp, primers: forward, 5'-CGTTTTCCATGATGCTCGGC-3' and reverse, 5'-CGAAGACCAACGCGATG-TAGTTCCA-3') were cloned into the EcoRI and NotI sites of *pSL1180-HR-PUbECFP* (Addgene #47917), respectively.

*AaegGr19* was targeted using *pSL1180-HR-PUBdsRED-Gr19*. The left homologous arm (1257 bp, primers: forward, 5'-AGGTATGCCTGGATTGGACGTAAGAAA-3' and reverse, 5'-GCAGTGAAAGG-TTGGTTAACTG-3') and right homologous arm (1203 bp, primers: forward, 5'-CCACCGGAAAGTG-GCATTTACCGC-3' and reverse, 5'-GCGACGGTCCCTTGCGATGTCGTTAT-3') were cloned into the XmaI and NotI sites of *pSL1180-HR-PUBdsRED* (Addgene #49327), respectively.

## Generation of mutant lines

To generate *AaegTRPA1* homologous recombination mutant alleles, 2000 pre-blastoderm stage wild-type embryos were microinjected (Genetic Services Inc., Cambridge, MA) with *AaegTRPA1* ZFN mRNA (200 ng/µl) and *pSL1180-HR-PUbECFP-TRPA1* (750 ng/µl). To generate *AaegGr19* homologous recombination mutant alleles, 1000 pre-blastoderm stage wild-type embryos were microinjected (IBBR Insect Transformation Facility, Rockville, MD) with *AaegGr19* ZFN mRNA (200 ng/µl) and *pSL1180-HR-PUBdsRED-Gr19* (700 ng/µl). Injected G0 animals were crossed to wild-type in multiple batches to generate G1 lines with independent ZFN mutagenesis events. G1 homologous recombination mutant individuals were recovered via fluorescence as previously described (*McMeniman et al., 2014*), and outcrossed separately to wild-type for five generations before establishing four independent homozygous lines: *AaegTRPA1*$^{ECFP-1}$, *AaegTRPA1*$^{ECFP-2}$, *AaegGr19*$^{DsRed-1}$, and *AaegGr19*$^{DsRed-2}$.

To confirm directed insertion of the *PUbECFPnls-SV40* cassette into the *AaegTRPA1* locus, a diagnostic PCR product (no wild-type band, 2075 bp mutant band) was amplified using a forward primer anchored outside the boundary of the left homologous arm (5'-CATGGGACAATTTGGCGTA-GGCAGTAT-3') and an ECFP reverse primer anchored inside the inserted cassette (5'- AGATCTCG-ACCCAAGAAAAAGCGGAAG-3'). To establish homozygous *AaegTRPA1*$^{-/-}$ lines, a diagnostic PCR product (679 bp wild-type band, 3305 bp mutant band) spanning the ZFN cut-site was amplified using primers: forward, 5-GGTTTCAAGGATGATTGACACACAAG-3', and reverse, 5'-GCAGAGCT-GATTTCTCGTAGTTTTCG-3'.

To confirm directed insertion of the *PUBdsRED-SV40* cassette into the *AaegGr19* locus, a diagnostic PCR product (6362 bp wild-type band, 8787 bp mutant band) was amplified using primers anchored outside the boundaries of both homologous arms: forward, 5'-AGCTGATCAACGTTAAC-AACTACGATG-3', and reverse, 5'-AGAGCATGGTGTAAACTTGACAGCTCAA-3'.

The following lines were used for all experiments: *AaegTRPA1*$^{ECFP-1/ECFP-2}$, *AaegTRPA1*$^{ECFP-1/+}$, *AaegGr19*$^{DsRed-1/DsRed-2}$, *AaegGr19*$^{DsRed-1/+}$. Heteroallelic mutants were used to minimize fitness effects that might be present in homozygous mutants.

## Behavior

All assays were carried out between ZT2-ZT12 at 26°C and 70–80% relative humidity unless stated otherwise. Whenever possible, time of day was randomized across conditions. All mosquitoes used were 10–21 day-old females, age-matched across conditions and genotypes.

### Heat-seeking

Assays were performed as previously described (*McMeniman et al., 2014*), in a 30 × 30 × 30 cm Plexiglas enclosure. All stimulus periods lasted 3 min and were presented on a single Peltier element (6 x 9 cm) covered with a piece of standard white letter size printer paper (extra bright, Navigator; Office Depot/Office Max, Cleveland, OH) cut to 15 × 17 cm and held taut by a magnetic frame. $CO_2$ pulses (20 s) accompanied all stimulus period onsets. A second identical control Peltier element was situated on the wall opposite to the stimulus Peltier and was set to ambient temperature during all experiments. Peltier temperature set-point is reported throughout the paper. Measurements of Peltier temperature via a thermocouple embedded in the Peltier element are reported in *Figure 1D*, *Figure 1—figure supplement 1A* and *Figure 2—figure supplement 1*. For each trial, 45–50 mosquitoes were introduced into the assay, and only mosquitoes directly on the Peltier area (*Figure 1B*) during seconds 90–180 of stimulus periods were scored. The custom MATLAB scripts used to run and analyze these experiments are available for download as *Source code 1 and 2* respectively. Heat maps are smoothed 2D histograms of mean mosquito occupancy during seconds 90–180 of stimulus periods, sampled at 1 Hz and binned into 12 × 16 image sectors.

### Heat-seeking choice

The assay was modified from the heat-seeking assay described above. For each trial, 45–50 mosquitoes were introduced into a custom-made Plexiglas box (38.5 × 28 × 12.5 cm) with two Peltier elements (4.6 × 6.5 cm surface area, ATP-O40-12, Custom Thermoelectric, Bishopville, MD) each covered with a taut piece of standard white letter size printer paper (extra bright, Navigator) cut to 9 × 11 cm. The Peltier elements were mounted on a single wall, 3.5 cm above the floor, 8.5 cm from either edge of the enclosure, and adjacent to one another separated by 6.5 cm. Mosquitoes were allowed to acclimate for 10 min, at which point the Peltier elements were both warmed to 40°C. After fully warming the Peltier elements for 60 s, a $CO_2$ pulse (20 s) was added to the airstream and the Peltier elements were kept at 40°C for a 4-min stimulus period at which point they were returned to ambient temperature (26°C). After a 12-min interstimulus period, the stimulus was repeated, this time with the left Peltier element warmed to 50°C while the right Peltier element was warmed to 40°C. A camera (Point Grey Research, Richmond, BC, Canada) was used to measure mosquito landings, and only mosquitoes directly on either Peltier area during seconds 60–240 of stimulus periods were scored. Preference was calculated by subtracting the proportion of total mosquitoes on the left Peltier (40 or 50°C) from the proportion of total mosquitoes on the right Peltier (40°C). Heat maps are smoothed 2D histograms of mean mosquito occupancy during seconds 60–240 of stimulus periods, sampled at 1 Hz and binned into 15 × 52 image sectors.

### Capillary feeder (CAFE)

This assay was adapted for the mosquito from the original assay developed for *Drosophila* (*Ja et al., 2007*) For each trial, five mosquitoes were fasted with access to water for 24 hr, and placed in a polypropylene vial (#89092–742, VWR, Radnor, PA) with access to two 5-μl calibrated glass capillaries (#53432–706, VWR) containing 10% (w: v) sucrose with 5% (v: v) green McCormick brand food dye. Capillaries spaced ~0.5 cm apart traversed the cotton plug (#49–101, Genesee Scientific, San Diego, CA) of the vial and protruded into the vial < 1 mm to provide a flush surface for mosquitoes to access the liquid while resting on the plug surface. The control capillary contained only green sucrose solution, and the experimental capillary contained green sucrose solution supplemented with 0, 1, or 10 mM of denatonium benzoate (Sigma-Aldrich) or 0, 10, 50, or 100 mM of N-methyl-maleimide (Sigma-Aldrich). The experimental capillary with 0 mM chemical was identical to the control capillary, and served as a zero-choice to test for side-bias in the assay. Because a small amount of liquid evaporated during preparation of the capillaries, all choice conditions and genotypes were prepared in a time-staggered format so that any measurement error due to evaporation was spread across the conditions. The levels of remaining liquid in both capillaries were measured after 18–20 hr

and were compared to the known initial liquid level. Experiments started at ZT 8–10 and ended at ZT 4–6 the following day. Consumption values were compared to control capillaries in vials without mosquitoes to account for evaporation. We note that in cases where mosquitoes did not feed from a capillary and all liquid loss was due to evaporation, consumption values could be calculated to be negative due to very small variation in evaporation rates between experimental and control capillaries. Any negative consumption values were rounded to zero. Sucrose preference was calculated by dividing the amount consumed from the control capillary (not containing denatonium benzoate or N-methylmaleimide) by the total amount consumed from both capillaries. In experiments with 0 mM denatonium benzoate or 0 mM N-methylmaleimide, one capillary was arbitrarily designated 'sucrose only'.

### Thermal gradient

This assay was adapted from one developed for *Drosophila* (*Sayeed and Benzer, 1996*; *Hamada et al., 2008*). A custom-built enclosure (6 mm tall) was affixed to an aluminum thermal gradient bar (50 × 30.5 cm, TGB-5030, ThermoElectric Cooling America Corp., Chicago, IL) driven by two Peltier elements (AHP-1200CPV, ThermoElectric Cooling America Corp., Chicago, IL). The enclosure was separated lengthwise into 4 lanes (each 50 × 6 cm) that were visually isolated from one another. Three lanes were for testing mosquito thermal preference, while the fourth lane was dedicated to measuring air temperature via an array of eight digital temperature sensors (DS18B20, Maxim, San Jose, CA; connected to an Arduino Uno, https://www.arduino.cc/) mounted to the top of the enclosure and distributed evenly across the length of the lane and centered in each analysis sector. An overhead camera (C910, Logitech, Lausanne, Switzerland) monitored mosquito position through the transparent lid of the enclosure. Images were acquired once per minute and analyzed using custom MATLAB scripts to count mosquitoes across eight analysis sectors of the lane. The MATLAB script is available for download as *Source code 3*. The assay was conducted in a room maintained at 80–90% relative humidity and 14°C to achieve low air temperatures within the gradient enclosure. At the beginning of each 3-hr trial, the air temperature throughout the enclosure was stabilized at ~26°C, and 25–30 mosquitoes were introduced into each lane of the assay. After 90 min, a thermal gradient was established (air temperatures: ~19°C to ~36°C) by heating the right Peltier element and cooling the left Peltier element. Mosquitoes were monitored for an additional 90 min while exposed to this thermal gradient. Mosquito distributions during minutes 60–90 were monitored in both the 'no thermal gradient' and 'thermal gradient' conditions. Dead mosquitoes were visually identified at the conclusion of each trial. All genotypes were tested in parallel, and their lane positions were randomized across trials.

## Thermal images

All thermal images were acquired with an infrared camera (E60, FLIR Systems, Wilsonville, OR).

## Statistical analysis

All statistical analyses were performed using Prism 5 software (GraphPad Software, Inc., La Jolla, CA).

## Acknowledgements

We thank Kevin Lee, Conor McMeniman, and members of the Vosshall Lab for discussions and comments on the manuscript; Nilay Yapici for advice, and Sarah-Yeoh Wang for assistance with the CAFE experiments in *Figure 3B–D*; Gloria Gordon for expert mosquito rearing; Conor J McMeniman for early work on the characterization of the *AaegTRPA1* locus and Benjamin J Matthews for *AaegGr19* RNA-seq analysis, both of which guided ZFN design; Lina Ni and Paul Garrity for valuable advice on the gradient assay in *Figure 3—figure supplement 1*; and Marco Gallio, Charles Zuker, and Paul Garrity for sharing unpublished information about *Drosophila* thermosensors prior to publication. This study was supported in part by a grant from the NIH CTSA program (NCATS UL1 TR000043). LBV is an investigator of the Howard Hughes Medical Institute.

## Additional information

### Funding

| Funder | Grant reference number | Author |
|---|---|---|
| Howard Hughes Medical Institute | Investigator | Leslie B Vosshall |
| National Center for Advancing Translational Sciences | UL1 TR000043 | Leslie B Vosshall |

The funders had no role in study design, data collection and interpretation, or the decision to submit the work for publication.

### Author contributions

RAC, Conception and design, Acquisition of data, Analysis and interpretation of data, Drafting or revising the article; LBV, Conception and design, Drafting or revising the article

### Author ORCIDs

Leslie B Vosshall, iD http://orcid.org/0000-0002-6060-8099

### Ethics

Human subjects: Blood-feeding procedures were approved and monitored by The Rockefeller University Institutional Review Board (protocol# LV-0652). Human volunteers gave their informed written consent to participate in mosquito blood-feeding procedures.

Animal experimentation: This study was performed in strict accordance with the recommendations in the Guide for the Care and Use of Laboratory Animals of the National Institutes of Health. All of the animals were handled according to approved institutional animal care and use committee (IACUC) protocols (#14756) of The Rockefeller University.

## Additional files

### Supplementary files

• Source code 1. Heat-seeking assay.

• Source code 2. Heat-seeking assay count.

• Source code 3. Gradient count.

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
