## [Decision Letter]

Thank you for submitting your work entitled "TRPA1 tunes mosquito thermotaxis to host temperatures" for consideration by *eLife*. Your article has been favorably assessed by a Senior Editor and three peer reviewers, one of whom is a member of our Board of Reviewing Editors. One of the three reviewers, Paul Garrity, has agreed to share his identity.

The reviewers have discussed the reviews with one another and the Reviewing Editor has drafted this decision to help you prepare a revised submission.

Summary:

This is an elegant manuscript that examines genes involved in thermosensation in *Ae. Aegypti* mosquitoes. The authors established a behavioral assay to test thermal preference and determined the range of temperatures that female mosquitoes find attractive. They then generated targeted mutations in TRPA1 and Gr19, previously shown to participate in *Drosophila* thermosensation, and examined their role in thermal preference. This is an important study because thermotaxis behaviors in mosquitoes have not yet been studied in depth (and quantitatively), and because the authors link a particular thermoreceptor to heat avoidance using genetic knockouts. The behavioral assays are quite nice and the analysis of mutant mosquitos impressive.

The following clarifications/additions would help improve the manuscript:

1) A figure showing the genomic organization of *TRPA1* and *Gr19* genes in wild type/mutant would be extremely useful to assess the genomic alterations.

2) For figures that plot occupancy on the Peltier, the mosquitoes tend to accumulate on the bottom of the Peltier. Since the differential between Peltier temp and ambient temp is relevant here, it would be valuable to know the temperature profile throughout the box (and particularly surrounding the edges of the Peltier). If the temp of the Peltier plate is uneven, then measuring the temp gradient across the plate could help the authors to better define the temp preference range for the mosquitoes. Alternatively, it might be possible to determine occupancy only in the central region of the Peltier, where the temperature is uniform.

3) Why is the occupancy in Figure 1 ~2-3x what is seen in Figure 3?

4) The authors suggest that the broad peak in temp preference (Figure 1) matches the range of host body temps (Figure 1), but shouldn't it also represent a trade-off between thermoregulatory (regulating internal body temp) and heat-seeking behavior? Can the authors comment on this trade-off? Do the authors have a suggestion for how these two processes could be uncoupled to study them separately in this system?

5) Figure 2 should be displayed as "% on Peltier" rather than "normalized% on Peltier". This is necessary to corroborate the claim that mosquitoes respond to the temperature differential not the absolute temperature, which would be very interesting.

6) Figure 3 explores the temporal dynamics of the temperature seeking behavior. However it is unclear whether the origin of the plots is at t=0 or some other time point. These plots indicate that in all cases (even at higher temperatures >45C) mosquitoes first land on the warm object and then leave it if it is too hot, or persist if it is around 40C. This implies that the "seeking" part of the behavior might only depend on detecting warm objects, and that the 90-180 second time window used for all quantifications in the paper comprises a combination of temperature seeking + heat avoidance behaviors. The *TrpA1^-/-^* mosquitoes seem to be defective in the heat avoidance behavior not the seeking behavior. These results/interpretations should be discussed in the text.

7) The Introduction does a good job of providing background on the mosquito thermotaxis literature, but could provide better background in comparing/contrasting the thermosensory system of the mosquito with the better studied *Drosophila* system. As warmth-sensing in *Drosophila* involves TRPA1 acting in internal thermosensors located in the brain and *Gr28b(D)* acting in peripheral thermoreceptors located in the antenna/arista, the authors may wish to comment on how this may relate (or not) to mosquito anatomy and what their findings (*Aedes* and *Drosophila* TRPA1s seem to have similar functions; Gr28b/Gr19 may not) may mean in this context.

8) Based on the data, statements about blood seeking do not seem relevant. The behavioral phenotype that the authors clearly demonstrated is decreased heat aversion rather than decreased attraction to humans. Statements about the role of *TrpA1* in host-seeking behavior of *A. aegypti* would be best left to the Discussion.

---

## [Author Response]

*The following clarifications/additions would help improve the manuscript:*

*1) A figure showing the genomic organization of* TRPA1 *and* Gr19 *genes in wild type/mutant would be extremely useful to assess the genomic alterations.*

We agree. This is now provided in Figure 3—figure supplement 1 (for TRPA1) and in Figure 3—figure supplement 2 (for Gr19).

*2) For figures that plot occupancy on the Peltier, the mosquitoes tend to accumulate on the bottom of the Peltier. Since the differential between Peltier temp and ambient temp is relevant here, it would be valuable to know the temperature profile throughout the box (and particularly surrounding the edges of the Peltier). If the temp of the Peltier plate is uneven, then measuring the temp gradient across the plate could help the authors to better define the temp preference range for the mosquitoes. Alternatively, it might be possible to determine occupancy only in the central region of the Peltier, where the temperature is uniform.*

The reviewers have identified one of the most important questions stemming from this work. We absolutely agree that it would be valuable to know more about the temperature that mosquitoes experience in this assay. This is a very difficult measurement to obtain and interpret, because thermal stimuli by their nature produce extremely complex and non-uniform thermal microenvironments. Moreover, to our knowledge there is no way to determine the instantaneous temperature that thermosensitive neurons in the mosquito sensory structures are experiencing. We could restrict our analysis to only report occupancy in the center of the Peltier, as was suggested, but we would still be unable to claim that these animals are experiencing uniform temperatures. We erred on the side of showing all the data because the behavior of the mosquitoes is likely driven by heat produced from the full surface area of the Peltier. To bring these ideas into the manuscript, we have added the following text to the Discussion:

“During interactions with a warm-blooded host or a 50°C Peltier, mosquitoes are likely experiencing a wide range of thermal fluctuations. […] Future studies characterizing this complex thermal microenvironment, and identifying the relevant thermosensory neurons and receptors will be required to define the thermal fluctuations experienced by mosquitoes during heat-seeking.”

*3) Why is the occupancy in Figure 1 ~2-3x what is seen in Figure 3?*

As in many behavioral assays in the laboratory, the absolute level of mosquito heat-seeking varies between experimental sessions. While all possible measures were taken to ensure consistency in mosquito rearing, we note that seasonal, meteorological, and other environmental effects can impact behavior. For this reason, we always include the essential controls within every series of experiments so that we can make appropriate comparisons.

*4) The authors suggest that the broad peak in temp preference (Figure 1) matches the range of host body temps (Figure 1), but shouldn't it also represent a trade-off between thermoregulatory (regulating internal body temp) and heat-seeking behavior? Can the authors comment on this trade-off? Do the authors have a suggestion for how these two processes could be uncoupled to study them separately in this system?*

This is a very important point, and we have expanded our discussion of these ideas in the context of our data. The following text has been added to the Discussion section:

“It may also be that these two thermotactic drives – host-seeking and thermoregulation – interact in mosquitoes. For instance, the avoidance of high-temperature stimuli during heat-seeking may be influenced by a nociceptive or thermoregulatory response independent of host-seeking behavior. It will be interesting to investigate the neural mechanisms that regulate divergent behavioral choices of thermoregulation and heat-seeking, and whether these systems are in behavioral conflict during mosquito host-seeking.”

*5) Figure 2 should be displayed as "% on Peltier" rather than "normalized% on Peltier". This is necessary to corroborate the claim that mosquitoes respond to the temperature differential not the absolute temperature, which would be very interesting.*

Due to the technical constraints of changing the ambient temperature in the environmental room where the behavior is carried out, the experiments in Figure 2 were conducted on different days, with different ambient temperature set points on each day. For this reason, we normalized these data to eliminate any effects due to variability in the absolute level of heat-seeking. We believe the important comparison is the relative amount of heat-seeking to different thermal stimuli in each thermal background. The raw, un-normalized data (Figure 2) also show that temperature differential is important for attracting mosquitoes (Figure 2), and that mosquitoes avoid the Peltier area for stimuli above ~55°C, regardless of the differential (Figure 2).

*6) Figure 3 explores the temporal dynamics of the temperature seeking behavior. However it is unclear whether the origin of the plots is at t=0 or some other time point. These plots indicate that in all cases (even at higher temperatures >45C) mosquitoes first land on the warm object and then leave it if it is too hot, or persist if it is around 40C. This implies that the "seeking" part of the behavior might only depend on detecting warm objects, and that the 90-180 second time window used for all quantifications in the paper comprises a combination of temperature seeking + heat avoidance behaviors. The* TrpA1^-/-^
*mosquitoes seem to be defective in the heat avoidance behavior not the seeking behavior. These results/interpretations should be discussed in the text.*

We thank the reviewers for pointing this out and have added text to the figure legend to clarify the time segments plotted in Figure 3. We have also modified the text (shown below) to more clearly point out the interesting dynamics seen in this figure. We certainly agree with the interpretation that TRPA1 is required for a specific component of thermotaxis in which high-temperature stimuli are avoided. We added the following sentence:

“Therefore, *AaegTRPA1* is not required for initial attraction to warmth, but is required for normal avoidance of high-temperature stimuli that exceed host body temperature.”

*7) The Introduction does a good job of providing background on the mosquito thermotaxis literature, but could provide better background in comparing/contrasting the thermosensory system of the mosquito with the better studied* Drosophila *system. As warmth-sensing in* Drosophila *involves TRPA1 acting in internal thermosensors located in the brain and* Gr28b(D) *acting in peripheral thermoreceptors located in the antenna/arista, the authors may wish to comment on how this may relate (or not) to mosquito anatomy and what their findings (*Aedes *and* Drosophila *TRPA1s seem to have similar functions; Gr28b/Gr19 may not) may mean in this context.*

This was a regrettable omission in our writing, and we agree that a more scholarly comparative review between mosquito and fly is important. The manuscript has been revised to include the following text:

In the Introduction:

“In *D. melanogaster*, TRPA1 is expressed in internal thermosensors located in the brain, and *DmelTRPA1^-/-^* mutants fail to avoid high air temperature in a thermal gradient (Hamada et al., 2008).”

In the Results:

“The different phenotypes of these mutations in *Ae. aegypti* and *D. melanogaster* may reflect differences in expression patterns of these genes. […] The cellular expression pattern of Gr19 in *Ae. aegypti* is not known, but its transcript is broadly expressed (Matthews et al., 2015).”

*8) Based on the data, statements about blood seeking do not seem relevant. The behavioral phenotype that the authors clearly demonstrated is decreased heat aversion rather than decreased attraction to humans. Statements about the role of* TrpA1 *in host-seeking behavior of* A. aegypti *would be best left to the Discussion.*

Good point. We have edited the manuscript to remove all speculation about the role of *AaegTRPA1* in the natural environment of host seeking. These ideas are now limited to the Discussion.